# Channel Estimation for UAV Communication Systems Using Deep Neural Networks

**Ahmed Al-Gburi** [1,†]**, Osamah Abdullah** [2,*,†]**, Akram Y. Sarhan** [3] **and Hayder Al-Hraishawi** [4]

1   Department of Computer Science, Al-Mustansiriyah University, Baghdad 10052, Iraq
2   Head of Computer Engineering Technique Department, Madenat Al-Elem University College,
    Baghdad 10006, Iraq
3   Department of Information Technology, College of Computing and Information Technology at Khulais,
    University of Jeddah, Jeddah 21959, Saudi Arabia
4   Interdisciplinary Centre for Security, Reliability and Trust (SnT), University of Luxembourg,
    L-1855 Luxembourg, Luxembourg
*   Correspondence: osamah.abdullah@wmich.edu
†   These authors contributed equally to this work.

**Abstract:** Channel modeling of unmanned aerial vehicles (UAVs) from wireless communications has gained great interest for rapid deployment in wireless communication. The UAV channel has its own distinctive characteristics compared to satellite and cellular networks. Many proposed techniques consider and formulate the channel modeling of UAVs as a classification problem, where the key is to extract the discriminative features of the UAV wireless signal. For this issue, we propose a framework of multiple Gaussian–Bernoulli restricted Boltzmann machines (GBRBM) for dimension reduction and pre-training utilization incorporated with an autoencoder-based deep neural network. The developed system used UAV measurements of a town's already existing commercial cellular network for training and validation. To evaluate the proposed approach, we run ray-tracing simulations in the program Remcom Wireless InSite at a distinct frequency of 28 GHz and used them for training and validation. The results demonstrate that the proposed method is accurate in channel acquisition for various UAV flying scenarios and outperforms the conventional DNNs.

**Keywords:** channel modeling; deep learning; optimization algorithms; unmanned aerial vehicles (UAVs)

## 1. Introduction

Low-altitude unmanned aerial vehicles (UAVs), also called drones, have enabled many personal and commercial applications, including aerial photography and sightseeing, parcel delivery, emergency rescue in natural disasters, monitoring and surveillance, and precision farming [1]. Recently, the interest in this emerging technology is steadily surging as many governments have already facilitated the regulations for UAV usage. As a result, UAV technologies are being developed and deployed at a very rapid pace around the world to offer fruitful business opportunities and new vertical markets [2]. In particular, UAVs can be employed as aerial communication platforms to enhance wireless connectivity for ground users and Internet of Things (IoT) devices in harsh environments when terrestrial networks are unreachable. Additionally, intelligent UAV platforms can provide important and diverse contributions to the evolution of smart cities by offering cost-efficient services ranging from environmental monitoring to traffic management.

Wireless communication is a key enabling technology for UAVs, and their integration has drawn substantial attention in recent years. In this direction, the third generation partnership project (3GPP) has been active in identifying the requirements, technologies, and protocols for aerial communications to enable networked UAVs in current long-term evolution (LTE) and 5G/B5G networks [3]. The communications of UAVs fundamentally differ from terrestrial communications in the underlying air-to-ground propagation channel

and the inherent size, weight, and power constraints. The 3D mobile UAVs enjoy a higher probability of line-of-sight (LoS) communication than ground users, which can be beneficial for the reliability and power efficiency of UAV communications. Nevertheless, this also implies that UAV communications may easily cause/suffer interference to/from terrestrial networks, which has to be carefully addressed [4]. The availability of LoS depends not only on the propagation environment but also on the UAVs' altitude, elevation angle, and movement trajectories, which have to be jointly evaluated for each scenario.

Realizing full-fledged UAVs in the 3D mobile cellular network depends largely on the reliability and efficiency of the communication channels over diverse UAV operating environments and scenarios. Furthermore, these channels are crucial for designing and evaluating UAV communication links for control/non-payload and payload data transmissions across novel UAV operation scenarios, which is one of the significant challenges in this setting. Moreover, the mobility of UAVs and the time-varying topology of the 3D mobile network, along with localization errors and latency, may complicate acquiring a timely and accurate knowledge of channel state information (CSI). Therefore, obtaining accurate channel modeling is paramount for designing robust and efficient beam-forming and beam-tracking algorithms, resource allocation methods, link adaptation approaches, and multiple antenna techniques. While several statistical air-to-ground channel models that trade off between accuracy and mathematical tractability have been proposed in the literature [5], more practical analysis to bridge this knowledge gap is still needed.

On a parallel avenue, the wireless communication community has paid significant attention to deep learning (DL) techniques owing to their success in various applications, e.g., computer vision, natural language processing, and automatic speech recognition. DL is a neuron-based machine learning approach that can construct deep neural networks (DNNs) with versatile structures based on the application requirements [6]. Specifically, several works in the open literature have utilized DL methods for channel modeling and CSI acquisition. For instance, a DL-driven channel modeling algorithm is developed in [7] using a dedicated neural network based on generative adversarial networks designed to learn the channel transition probabilities from receiver observations. Furthermore, in [8], a DNN-based channel estimation scheme is proposed to jointly design the pilot signals and channel estimator for wideband massive multiple-input multiple-output (MIMO) systems.

To address the limitations of existing systems, researchers have focused on ML and DL methods in recent years. Despite their enormous potential, DL approaches confront significant difficulties that limit their applicability in advanced communication environments. To develop an adequate mapping from features to desired outcomes, deep NNs (DNNs) require enormous datasets, which raises both the computational complexity and the difficulty of the training process. Due to the current state of communication datasets, it is questionable to rely purely on data-based DNNs as black boxes and leave all predictions to model weights. Model-driven DNNs have evolved as a solution to the limitations of model-based and data-based techniques; these DNNs combine the strengths of DNNs in learning and mapping with domain expertise to maximize benefits [9]. Most model-driven DNN solutions fall into one of two categories: deep unfolded networks, in which DNN layers reproduce rounds of an existing iterative process, or hybrid networks, in which DNNs assist conventional models and boost efficiency . The data scarcity problem has also been eliminated alongside the advancement of DL literature on communication technologies, making room for data-based DL systems [10]. Nevertheless, the promising enhancements of DL techniques for channel modeling have motivated researchers to investigate utilizing different learning methods and feature extraction approaches in the context of cellular-connected UAV communication systems. To the best of our knowledge, only a few prior relevant research works dealt with UAV channel characterizations using the received signal strength (RSS) in cellular communications [11–14]. In [11], a modeling framework for wave propagation in mobile communications is proposed by combining several learners in an ensemble learning method for RSS modeling. Furthermore, a deep reinforcement learning (DRL) -based model for channel and power assignment is developed in [10] for

UAV-enabled IoT systems, where a single UAV-base station is deployed to collect data from multiple IoT nodes. In [13], a learning algorithm is used to predict channel characteristics between UAVs and ground users, which provides accurate environmental status information for UAV deployment decisions. Additionally, the authors in [14] have proposed ensemble methods based on supervised base learning to predict the channel model of UAV using the least squares boosting method, bagging prediction, and support vector machines (SVMs).

In terms of DL methods, Wang et al. [15] have proposed a DL scheme that can fully explore the features of wireless channel data and obtain the optimal weights as fingerprints while incorporating a greedy learning algorithm to reduce computational complexity. Beyond this, the restricted Boltzmann machine (RBM) is a generative stochastic artificial neural network that can learn a probability distribution over a set of inputs in an unsupervised manner. RBM is bipartite, i.e., there are no intra-layer connections, and it consists of a pair of layers that are commonly referred to as the *visible* and *hidden* units, respectively, as shown in Figure 1, and they may have a symmetric connection between them. However, RBM has some limitations due to dealing with only binary data, and to circumvent this issue, the Gaussian–Bernoulli restricted Boltzmann machine (GRRBM) [16] was proposed to process real data where a Gaussian visible node replaced a binary node to initialize DNN for feature extraction and dimension reduction.

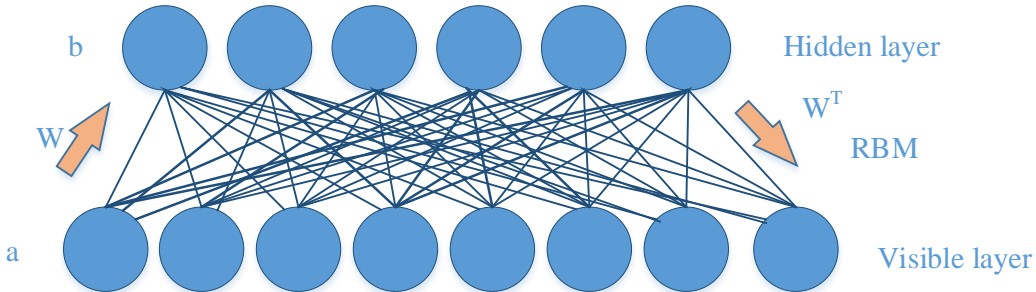

**Figure 1.** The network of RBM diagram.

The commonality of the works as mentioned earlier [11–15] is that they all follow training methods that have high variance and slow diverging behaviors that require a massive volume of training data and extended training time, which is prohibitive in wireless communication systems. This observation has motivated this work to fill this gap in the literature and propose a framework based on GBRBM that incorporates an autoencoder-based deep neural network to estimate the received signal power of UAVs flying in a range of altitudes connected to a cellular network. Moreover, we propose a novel algorithm that employs an adaptive learning rate alongside an enhanced gradient, which speeds up the learning of hidden neurons, contrary to the traditional gradient decent. The main contributions of this work can be summarized as follows:

- Apply GBRBM and real measurements to estimate the received signal power at UAV from the cellular network during the flight, where GBRB machines and deep neural network are employed to extract features from the UAV channels as a series set of blocks for channel modeling.
- Develop an adaptive learning rate approach and a new enhanced gradient to improve the training performance. Specifically, an autoencoder is used to fine-tune the parameter, while the parameters are trained by using an encoder neural network to model the RSS for the prediction of the ground at different heights.
- Verify the effectiveness of the proposed method throughout experimental measurements and comparisons with other benchmark schemes. The numeral results show that the proposed scheme outperforms the conventional autoencoders.

The remainder of this paper is organized as follows. Section 2 presents the system model. The proposed method is described in Section 3. Numerical examples and demon-

strations of various simulation results are given in Section 4. Finally, conclusions are drawn in Section 5.

## 2. System Model and Problem Formulation

Communications and transportation technologies are major assets for our lifestyle. In the move toward the integration of these advancements in one field, a transceiver can be attached to a drone to assist future wireless networks. Significantly, in natural disasters, especially when communications infrastructure (e.g., base stations) is damaged, the communication network must be maintained for search and rescue efforts. Currently, existing methods have limitations in terms of flexibility and resource availability. Coping with such issues requires that unmanned aerial vehicles (UAVs) be employed to work wirelessly because of their multifunctional qualities and flexibility. A drone flies at low altitude and is equipped with transceivers to work with a wireless network as a drone cell in the coverage area. Many parameters determine the channel modeling between the drone transceiver and the BSs, such as the altitude, antenna directivity, location, transmission power, and the characteristics of the environment. To investigate the effects of these parameters on channel modeling between the drone transceiver and the BSs, we proposed a framework of GBRBMs for dimension reduction and pre-training utilization incorporated with an autoencoder-based deep neural network.

### 2.1. Problem Formulation

GBRBM is known as Markov random fields (MRFs) [17] and is an undirected probabilistic graphical model. The input layer represents the observed data consisting of nine nodes, each of which represents a specific kind of input data, N = (N1 − N9), where N1 represents the latitude, N2 represents the longitude, N3 is the drone ground elevation, N4 and N5 represent the cell latitude and longitude, respectively, n6 represents the cell elevation, N7 is the cell building, N8 is the antenna mast height, and N9 represents the drone altitude. Figure 2 illustrates the diagram enhanced GBRBM with DNN. This architecture is designed to operate systematically on the principle of the proposed algorithm. First, data are collected during the real-time data collecting stage. Next, the raw data will be sent to the pre-training stage. Then, the data are exploited for feature extraction using multi-block enhanced GBRBM; next, we send the pre-trained data to the fine-grained tuning. Finally, the output data of the extracted features are classified using the softmax regression module to obtain the output.

The energy function of GBRBM is defined as:

$$E(v, h|\theta) = \sum_{i=1}^{n_v} \frac{(v_i - b_i)^2}{2\sigma^2} - \sum_{i=1}^{n_v} \sum_{j=1}^{n_h} W_{ij} h_j \frac{v_i}{\sigma_i} - \sum_{j=1}^{n_h} c_j h_j \tag{1}$$

where $b_i$ represents the bias of the visible layer, $c_j$ represents the bias of the hidden layer, and $w_{ij}$ represents the weight that connects the visible layer to the hidden layer. Furthermore, $\sigma_i$ accounts for the standard deviation of the visible units. Each block has nine real input data, as mentioned above, while $n > 1$ for hidden layer unit $h_j$. Based on the property of MRFs and the energy function, the joint probability distribution is defined as:

$$P(v, h) = \frac{1}{Z} e^{-E(v,h)}, \tag{2}$$

where Z is the partitioning function, as given below

$$Z = \sum_v \sum_v -E(v, h), \tag{3}$$

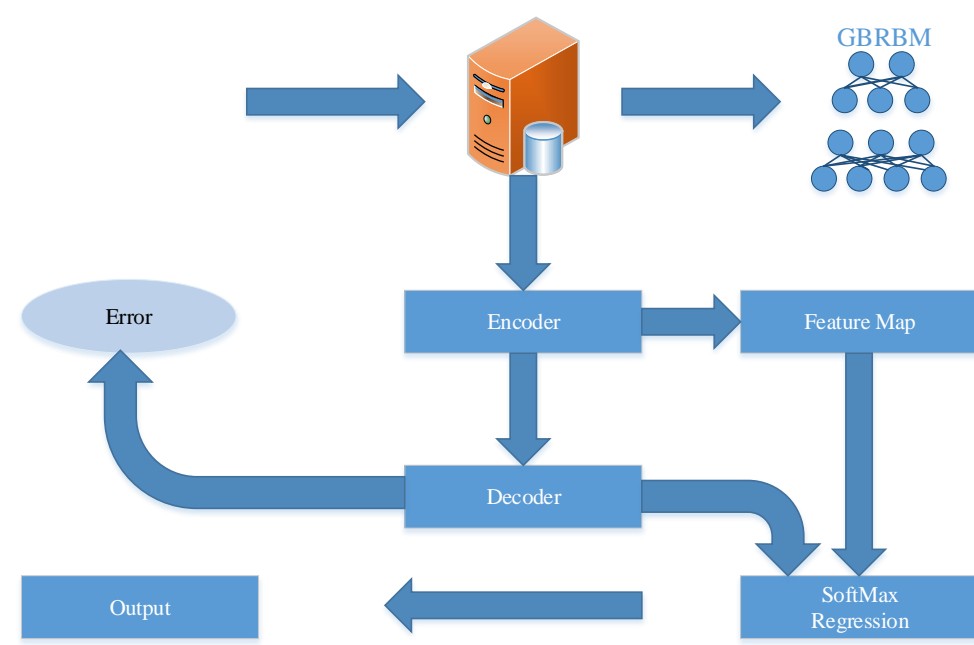

**Figure 2.** Framework architecture of the proposed enhanced GBRBM-based DNN.

By using the joint probability function, the marginal distribution of v can be defined as:

$$P(v) = \frac{1}{Z}e^{-E(v,h)},\tag{4}$$

The hidden and visible layers are both conditionally independent. The condition probability of $v$ and $h$ are defined as follows:

$$P(v_i = v|h) = N\left(v|b_i + \sum_J h_j W_{ij}\sigma_i^2\right),\tag{5}$$

$$P(h_j = 1|v) = \text{sigmoid}\left(c_j + \sum_i W_{ij}\frac{v_i}{\sigma_i^2}\right),\tag{6}$$

where $N$ represents the Gaussian probability density function and denotes the variance with mean. The stochastic maximization of likelihood is used to train GDBM. The likelihood is estimated by marginalizing the hidden neurons. The partial derivative of the maximization log-likelihood function is given by:

$$\frac{\partial L}{\partial\theta}\alpha\left\langle\frac{\partial\left(-E\left(v^{(t)},h|\theta\right)\right)}{\partial\theta}\right\rangle_d - \left\langle\frac{\partial(-E(v,h|\theta))}{\partial\theta}\right\rangle_m\tag{7}$$

where $\langle.\rangle_d$ and $\langle.\rangle_m$ account for the expectation of model distribution $P(v,h|\theta)$ and data distribution $P(h|\{v^{(t)}\},\theta)$, respectively. Here, $\theta$ is the parameter of GBRBM because gradient calculation needs a high computational cost. Reference [18] used a contrastive-divergence (CD) learning that proved to be an efficient approximator for the log-likelihood gradient for GBRBM. The CD learning is recalled as an alternative to calculating the second term of the log-likelihood gradient by iteration a few samples from the data by using Gibbs sampling. As a result, GBRBM parameters are derived as follows:

$$W_{ij} \quad \overset{=}{\Longleftarrow} \quad W_{ij} + \eta \left( \left\langle \frac{1}{\sigma_i^2} v_i h_j \right\rangle_d - \left\langle \frac{1}{\sigma_i^2} v_i h_j \right\rangle_m \right), \tag{8a}$$

$$b_i \quad \overset{=}{\Longleftarrow} \quad b_i + \eta \left( \left\langle \frac{1}{\sigma_i^2} v_i \right\rangle_d - \left\langle \frac{1}{\sigma_i^2} v_i \right\rangle_m \right), \tag{8b}$$

$$c_j \quad \overset{=}{\Longleftarrow} \quad c_j + \eta \left( \left\langle \frac{1}{\sigma_i^2} h_j \right\rangle_d - \left\langle \frac{1}{\sigma_i^2} h_j \right\rangle_m \right), \tag{8c}$$

where $\eta$ denotes the learning rate. The GBRBM is updated and trained efficiently by updating (8a)–(8c).

The RBM has been used as a universal approximator. Nevertheless, the number of hidden nodes is always limited, which leads to the inability to model some discrete-valued probability distribution [19]. In practice, it has been proved that not all the hidden nodes are active, and some are purposeless, where the weight associated with $m$ cannot be incorporated into the bias term. We introduce the adaptive learning rate and a new enhanced gradient in the next part.

### 2.1.1. Adaptive Learning Rate

Based on the maximization of the local estimate of the likelihood, the learning rate can automatically be adapted while the RBM is trained using the stochastic gradient. Because $\theta = (W, b, c)$ is the parameter of GBRM modeling, and it represents the adapted parameter of learning rate $\eta$, $P_\theta(V) = P_\theta^* / Z_\theta$ represents the probability density function (pdf), and $Z_\theta$ is the normalization parameter for the parameter $\theta$. If the difference between the learning rate becomes small and the models become close to each other, then the likelihood of $\theta'$ is computed as:

$$\begin{aligned} P_{\theta'}(v_d) &= \frac{P_{\theta'}^*(v_d)}{Z_\theta} \frac{Z_\theta}{Z_{\theta'}} \\ &= \frac{P_{\theta'}^*(v_d)}{Z_\theta} \left\langle \frac{P_{\theta'}^*(v)}{P_{\theta'}^*(v)} \right\rangle_{P_\theta}^{-1} \end{aligned} \tag{9}$$

Now, the learning rate parameters are chosen to maximize the likelihood of the GBRPM parameters $\theta'$. $P_{\theta'}$ represents the unnormalized computed PDF of the training data, which is used to estimate the expectation of the data. The optimal learning rate can be found that can maximize the likelihood of each iteration. Nevertheless, this can lead to a big fluctuation due to the small size of the minibatch. Ref. [20] proposed that the new learning rate is chosen from the set $\left\{ (1 - \mathcal{E})^2 \eta_o, (1 - \mathcal{E}) \eta_o, \eta_o, (1 - \mathcal{E}) \eta_o, (1 - \mathcal{E}) \eta_o, \right\}$ where $\eta_o$ is the prior learning rate and $\mathcal{E}$ is a small constant, which was chosen randomly.

### 2.1.2. Enhanced Gradient

The recently enhanced gradient was proposed by [21] to update the invariant rule of the Boltzmann machines for data representation. A bit-flipping transformation introduced the gradient, and then, the rule was updated to improve the results, the learning of the RBM, and make it less sensitive to the parameter variation and initialization. The same idea was used to enhance the gradient of GDBM by using Gaussian visible neurons as an alternative to the bit-flipping transformations by shifting the visible unit. Ref. [21] proposed a new method to enhance the gradient to replace the equations of (8a)–(8c). They defined the covariance between the two variables under the distribution $P$

$$COV_P(v_i, h_j) = \langle v_i h_j \rangle_P \langle v_i \rangle_P \langle h_j \rangle_P \tag{10}$$

The standard gradient in (8a) can be rewritten as

$$\begin{aligned} \nabla_{W_{ij}} &= COV_d(v_i, h_j) - COV_m(v_i, h_j) \\ &+ \langle v_i \rangle_{dm} \nabla_{c_j} + \langle h_j \rangle_{dm} \nabla_{b_i}, \end{aligned} \tag{11}$$

where $\langle . \rangle_{dm} = \frac{1}{2} \langle . \rangle_d + \frac{1}{2} \langle . \rangle_m$ denotes the average of the model distribution with the data. The standard deviation has some potential problems. The gradient is correlated with the weights and the bias terms. Additionally, $COV_d(v_i, h_j) - COV_m(v_i, h_j)$ is uncorrelated with $\nabla_{b_i}$ and $\nabla_{c_j}$, which may lead to distract the learning with non-useful weights when there are a lot of active neurons for which $\langle . \rangle_{dm} \approx 1$. The current problem is that using (11) to update the data varies if we depend on data representation, and that variation can make the data flip from one to zero and vice versa

$$\begin{aligned} \tilde{v}_i &= \tilde{v}^{(1-f_i)}(1-v_i)^{f_i} & f_i \in \{0,1\}, \tag{12a} \\ \tilde{h}_i &= \tilde{h}^{(1-g_i)}(1-h_i)^{g_j} & g_i \in \{0,1\}, \tag{12b} \end{aligned}$$

The parameters are transformed accordingly to $\tilde{\theta}$

$$\tilde{w}_{ij} = (-1)^{f_i + g_j} w_{ij}, \tag{13a}$$

$$\tilde{b}_i = (-1)^{f_i} \left( b_i + \sum_j g_j W_{ij} \right), \tag{13b}$$

The energy function is equivalent to $E(\tilde{x} + \tilde{\theta}) = E(x + \theta) + a$, where $a$ is a constant for all the values. That will lead to an update of the model and then transformed again. The resulting model will be defined as

$$\begin{aligned} w_{ij} \leftarrow w_{ij} + \eta \big[ COV_d(v_i, h_j) - COV_m(v_i, h_j) \\ + (\langle v_i \rangle_{dm} - f_i) \nabla c_j + \left( \langle h_j \rangle_{dm} - g_j \right) \nabla_{b_i} \big], \end{aligned} \tag{14a}$$

$$b_i \leftarrow b_i + \eta \left[ \nabla b_i - \sum_i g_j (\nabla w_{ij} - f_i \nabla c_j - g_i \nabla b_i) \right], \tag{14b}$$

$$c_j \leftarrow c_j + \eta \left[ \nabla c_j - \sum_i f_i (\nabla w_{ij} - f_i \nabla c_j - g_i \nabla b_i) \right] \tag{14c}$$

where $\nabla \theta$ represents the gradient parameters defined in Equation (8). Therefore, it will be a $2^{nv+nh}$ different update rule, where $nv$ and $nh$ are the numbers of hidden and visible neurons. In order to find the maximum likelihood updates, Ref. [20] proposed a new gradient weighted sum $2^{nv+nh}$ with the following weights:

$$\prod_i \langle v_i \rangle_{dm}^{f_i} (1 - \langle v_i \rangle)^{1-f_i} \prod_j \langle h_j \rangle_{dm}^{g_j} (1 - \langle h_i \rangle)^{1-g_j} \tag{15}$$

Due to the larger weights, the enhanced gradient is defined as:

$$\nabla_e w_{ij} = COV_d(v_i, h_j) - COV_m(v_i, h_j), \tag{16a}$$

$$\nabla_e b_i = \nabla b_i - \sum_j \langle h_j \rangle_{dm} \left( \nabla W_{ij} - \nabla b_i - \langle v_i \rangle_{dm} \nabla c_j \right), \tag{16b}$$

$$\nabla_e c_j = \nabla c_j - \sum_i \langle v_i \rangle_{dm} \left( \nabla W_{ij} - \nabla c_j - \langle h_j \rangle_{dm} \nabla c_j \right). \tag{16c}$$

where $\nabla_e w_{ij}$ has the same form of (10) and the bias gradient terms is deleted, as shown in Figure 3.

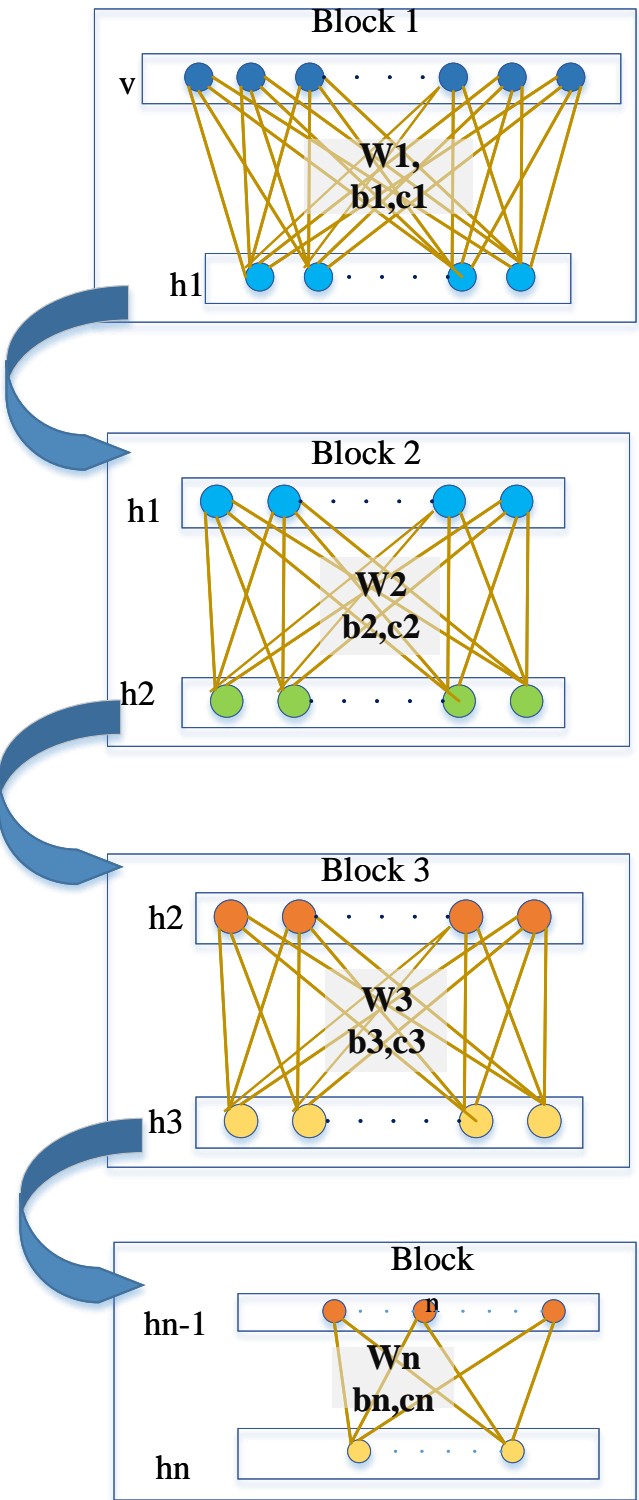

**Figure 3.** GBRBM blocks model.

Where each block is connected to the upper block through the unit of the hidden lower to update the parameter in (16) layer by layer for GBRBM pre-training. The pre-training of GBRBMs provides the initializing to deep autoencoder. The autoencoder is used to

reconstruct the input data that do not have a class label. The output of the autoencoder is defined as follows:

$$e(v) = f\left(c_j + \sum_i w_{ij}\frac{v_i}{\sigma_i^2}\right) \tag{17}$$

The deep autoencoder neural network architecture is depicted in Figure 4. The deep autoencoder, as seen in Figure 4, consists of an encoder phase and a decoder phase. The number of neurons for each layer of the network must be determined after the number of layers is chosen. Unfortunately, the optimization process is quite time-consuming because there is no set range for the number of neurons in each layer. It should be noted that the bottleneck layer includes the neural network's most fundamental input properties, and that the performance of the neural network depends on these features. In other words, the bottleneck layer's neuron count needs to be precise, the size of the bottleneck layer exceeds 8, and the detection accuracy is becoming saturated. As a result, in the DNN core architecture, the fifth network layer's neuron count is set to 8, where both the encoder and the decoder share a similar structure. Once the hidden layers have been unrolled, the decoded output can be reconstructed as:

$$r(v) = f\left(c_j' + \sum_i \sigma_i^2 w_{ij}e(v)\right) \tag{18}$$

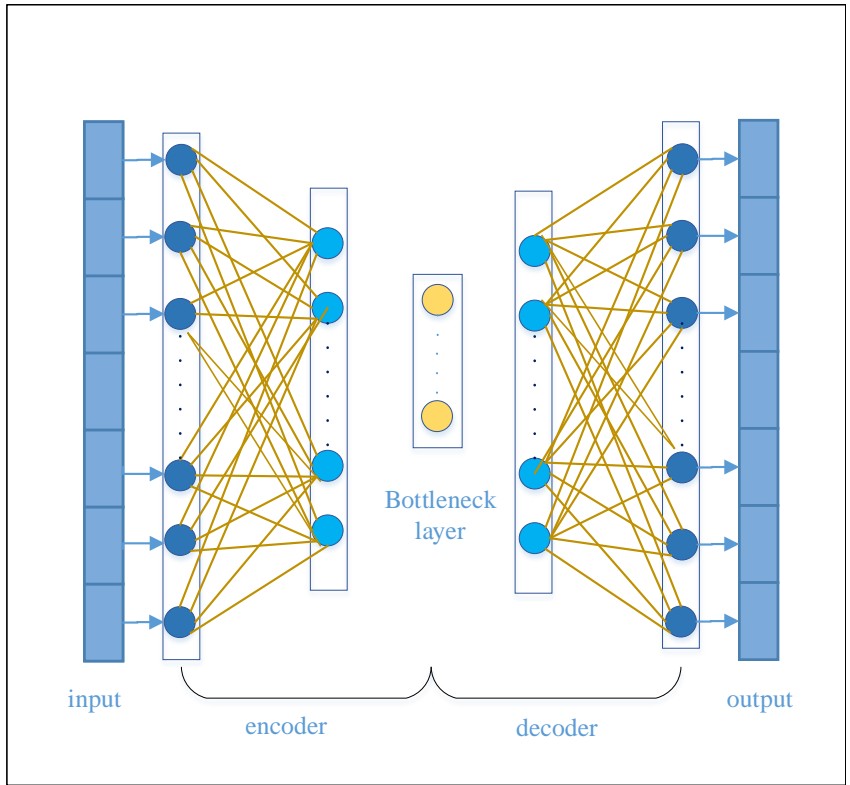

**Figure 4.** Structure of Encoder and Decoder in Deep Neural Network.

After that, the mean-square-error (MSE) cost function is used in the fine-grained stage to optimize the proposed algorithm through the backpropagation algorithm:

$$Error(D) = \frac{1}{N}\sum_{i=1}^{N}(r(v) - v)^2 \tag{19}$$

where $N$ represents the sample number of data input. Autoencoder training is unsupervised since it seeks to recreate the input data without any reference to the class labels. The final layer of the encoder network, depicted in Figure 4, is referred to as a bottleneck layer, and a softmax classifier is put behind it after features have been extracted using GBRBMs and an autoencoder. In particular, we use the softmax classifier to assess the likelihood that the input belongs to specific classes. Ref. [21] proposed a loss function defined as:

$$L(w_o, c_o) = \frac{1}{N} y_i \log\big(y_i'(w_o, c_o)\big) \qquad (20)$$

where $w_o$ and $c_o$ denote the encoder parameters and $y$ and $y_0$ represent the predicted and real labels. Equation (20) represents the loss function using the enhanced gradient algorithm with the backpropagation algorithm as shown in Algorithms 1–3.

---

**Algorithm 1:** Pre-training the GBRBM blocks (Unsupervised learning)

**Input:** Epoch number, $v$, $K$, GBRBM number, $L$
**Output:** $\nabla_o w_{ij}, \nabla_o c_j$

1   Initialize $\nabla w_{ij}, \nabla b_i, \nabla c_j$ randomly, $\mathbf{v}_{dm}^1 \leftarrow \mathbf{v}$
2   **while** *for each epoch* **do**
3     **while** $m \leq M$ **do**
4       Calculate $h_{dm}^m$ from (5)
5       Calculate $v_{dm}^m$ from (6)
6       Update the value of $\nabla w_{ij}, \nabla b_i, \nabla c_j$ using (16)
7     **end**
8     Repeat until the convergence is met
9   **end**

---

**Algorithm 2:** Fine tuning of autoencoder (Unsupervised Learning)

**Input:** Epoch number, $v$, $K$, GBRBM number, $L$
**Output:** $\nabla_o w_{ij}, \nabla_o c_j$

1   Initialize $\nabla_e w_{ij} \leftarrow \nabla w_{ij}, \nabla_e b_i \leftarrow \nabla b_i, \nabla_e c_j \leftarrow \nabla c_j$
2   **while** *for each epoch* **do**
3     **while** $m \leq M$ **do**
4       Estimate the decoded output using (18)
5     **end**
6     Estimate the MSE using (10)
7     Fine tune $\nabla_e w_{ij}, \nabla_e b_i, \nabla_e c_j$ using backpropagation
8     Repeat until the convergence is met
9   **end**

---

**Algorithm 3:** Fine tuning of the data labels (Supervised Learning)

**Input:** epoch number, $v$, $K$, GBRBM number, $L$
**Output:** $\nabla_o w_{ij}, \nabla_o c_j$

1   Initialize $\nabla_o w \leftarrow \nabla_e w^m, \nabla_o b \leftarrow \nabla_e b^m, \nabla_o c \leftarrow \nabla_e c^m$
2   **while** *for each epoch* **do**
3     Use backpropagation to estimate the fine tuning $\nabla_o c, \nabla_o w$
4     Repeat until the convergence is met
5   **end**

### 3. UAV Measurements

In this section, simulation results are provided to evaluate the performance of the proposed CSI prediction scheme for air-ground links in UAV communication systems. For UAV communications, mmWave bands hold a lot of promise to meet the data rate requirements of high-throughput mobile applications. Particularly, by hovering at a preferable place, UAVs can keep line-of-sight (LOS) connection (or at least an acceptable NLOS) link with a desired user. Ray tracing offers a deterministic method of characterizing the mmWave channel under various conditions.

Since studying the behavior of Air-to-Ground channels at mmWave bands using UAVs might be difficult, ray tracing provides a useful alternative. Here, we utilize the Remcom Wireless InSite ray tracing program to simulate the UAV's real-time movements along a specified trajectory. The simulation parameters were set up as follows: the area's dimensions are 10 km by 10 km in size as shown in Table 1. In all situations, the UAVs are flying at a speed of 15 m per second from altitudes of 200 m (resembling a land vehicle); the flight trajectory of the UAV is around 2 km in length. Both the transmitter and receiver employ vertically polarized half-wave dipole antennas. The channel at 28 GHz has a sinusoid to sound the channel at the center of the ferquency. The power level is set at 30 dBm for transmission [22].

**Table 1.** Simulation parameters.

| Scenario | Building Heights (m) | Number of Buildings |
|----------|----------------------|---------------------|
| Urban    | 100–120              | 100                 |
| Suburban | 20–30                | 25                  |
| Rural    | 5–10                 | 10                  |
| Overseas | -                    | -                   |

Considering mmWave frequencies of 28 GHz, we present the cumulative distribution functions (CDFs) of the RMS-DS of the multipath channel between the GS transmitter and the UAV in Figure 5 for four different scenarios. The major reason for this behavior is that at higher UAV altitudes, the UAV moves above tall structures and is able to observe signals that are scattered from a majority of surrounding buildings. In contrast, findings in rural and suburban settings demonstrate that unlike in urban settings, the RMS-DS actually declines as the number of buildings increases. Buildings in rural and suburban areas tend to be smaller and less densely populated than those in cities. These various multipath channel behaviors imply that the environmental variables and UAV height may have a big impact on the channel behavior and consequently the receiver design.

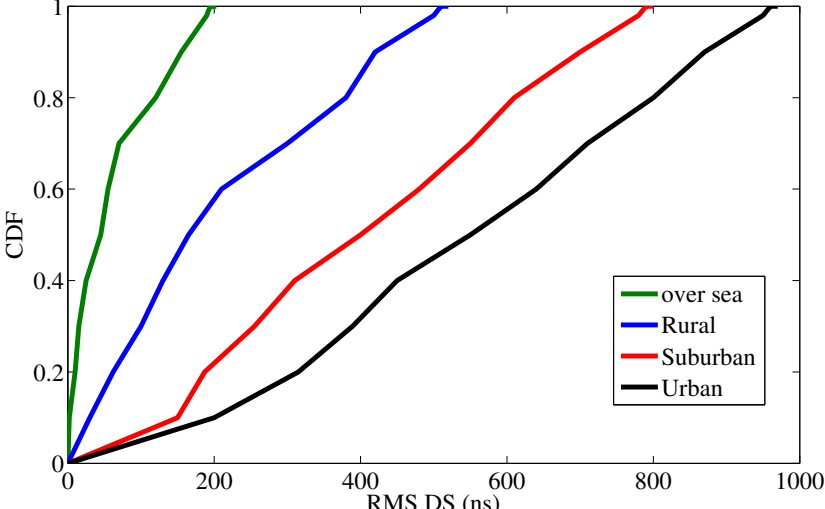

**Figure 5.** CDF of RMS-DS for different scenarios at 28 GHz .

## 4. Performance Evaluation

In this section, we implement the proposed GBRBM-based DNN in addition to other machine learning algorithms, i.e., backpropagation ANN and SVM. The considered simulation parameters include the GBRBM block, the number of neurons and layers of the network, and the value of the adaptive learning rate and number of the epoch. The total number of training vectors is 710, while the number of test vectors is 177, with 201 unknown variables with 100 iterations.

### 4.1. GBRBM Blocks

Figure 6 shows the difference error between the measurements and the estimated values by using different numbers of GBRBM-based DNN. Different numbers of GBRBM are considered starting from 2 to 7, while the differences error between the estimated and measured RSS varies from −5 to 15. It can be readily seen that the best results are obtained when six blocks of GBRBM-based DNN in the pre-training stage are used. The performance accuracies are 85.1%, 87.3%, 90.1%, 92.8%, 94.1%, and 93.7%, respectively.

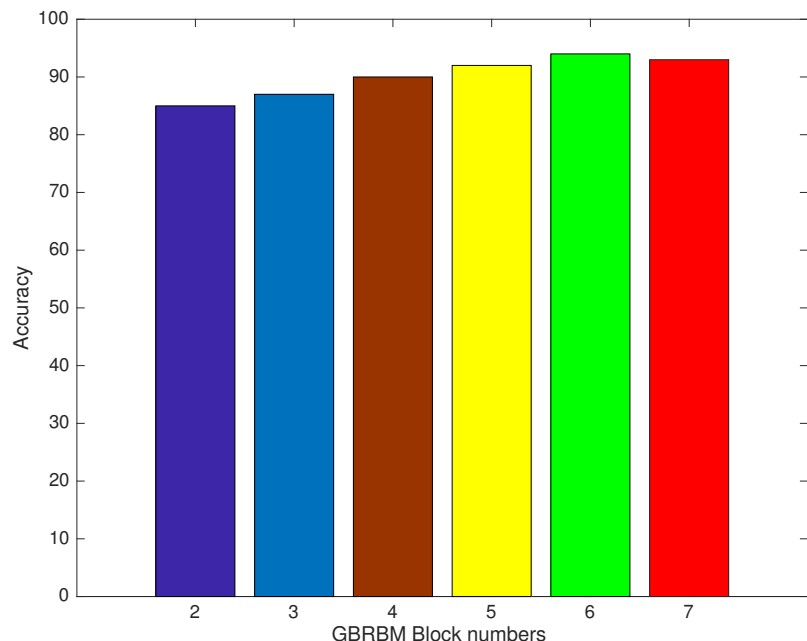

**Figure 6.** GBRBM-based DNN in the pre-training stage versus the accuracy.

Since a proper epoch number can shorten the training period and the search space of solutions, the best epoch number is important to both the pre-training and training phases. Therefore, the perfect epoch number could be selected based on the value of this reduction function. For example, Figure 7 presents the difference error versus the number of epochs. Clearly, the difference error is stable when we reached around the 500th epoch. Hence, the epoch number of the pre-training stage is set to 250, and the epoch number of the training stage is set to 500 epochs.

After the GBRBM blocks and epoch number are determined, the neuron number must be set for every layer. The optimization process is quite tedious because the tuning assortment of the neuron number in every layer is arbitrary. Thus, we empirically set the neuron amounts and then run experiments to maximize the neuron amount of the sixth layer depending on the operation of both GBRBM-based DNN. In the pre-training stage, the difference in the RSS value between the estimated measure and the real measure decreases as the epoch number increases and is set to the 250th epoch, as depicted in Figure 8.

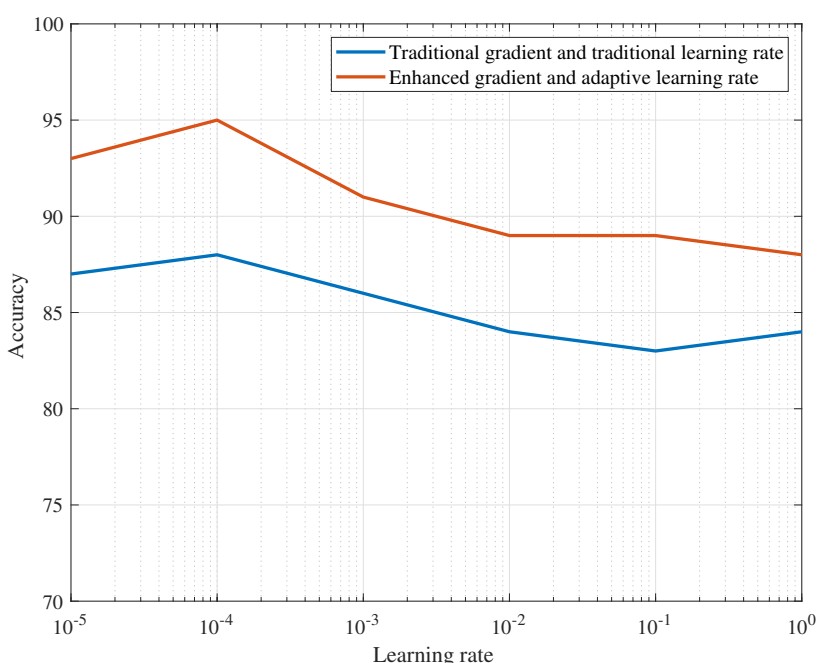

**Figure 7.** Accuracy versus adaptive learning rate.

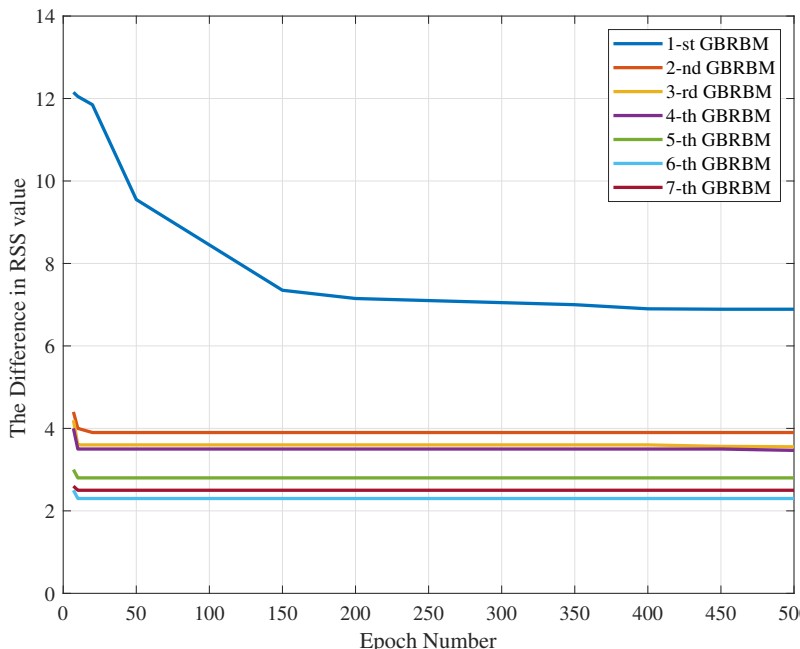

**Figure 8.** The average difference between the estimated and real RSS values in the pre-training stage.

### 4.2. Adaptive Learning Rate

The learning rate is similar to the step size of the gradient descent process. The precision will be greatly affected if it is set too big or too small. In particular, if the learning rate is too small, the training period grows, and a local optimal solution is likely to be trapped. Since there are pre-training and training stages, we must discover an individual optimum learning rate for all of them. We trained the traditional gradient with five learning rates (1, 0.1, 0.01, 0.001, 0.0001) to demonstrate how the learning speed can greatly affect training results. The resulting RBMs have enormous variance determined by the learning rate selection. When the learning rate was large, the result models completely failed, whereas better

results were obtained when the learning rate was too low. To test the suggested adaptive learning rate, we trained RBMs of the hidden neurons with the traditional gradient and the same five values (1, 0.1, 0.01, 0.001, 0.0001) to initialize the learning rate. Of this test information, the outcomes are more stable, the variance among the consequent RBMs is smaller than the results obtained with the learning rate regardless of the initial learning rate, and all RBMs were trained efficiently. These results have revealed that the adaptive learning rate performs better, leading to ameliorated results. However, it was slightly better to use a continuous learning rate of 0.001 in both pre-training and training stages.

Figure 7 shows the adaptive learning rate performance during learning. The process found suitable learning rate values when the enhanced gradient was used. Specifically, one will find six GBRBM blocks for the pre-training stages and five network layers to the training in the autoencoder. The neuron numbers for hidden layers of multi-block GBRBM are 64, 56, 48, 32, and 16, respectively. The epoch amounts of the pre-training and coaching phases are set to 250 and 500, respectively. The learning speeds of the two phases are equally set to 0.001, as shown in Table 2.

**Table 2.** Simulation parameters.

| Key Parameters | Settings |
|---|---|
| Number of layers | 5 |
| Neuron number of bottleneck layer | 7 |
| Number of GBRBMs | 5 |
| Epoch number of pre-training stage | 250 |
| Epoch number of training stage | 500 |
| Learning rate | 0.001 |

The GBRBM-AE parameters were set as training and pre-training algorithms to compare the results for 50 independent trails to obtain the mean, best, and worst values, as shown in Table 3.

**Table 3.** Comparative results using different training algorithm methods .

| Algorithm | Best | Worst | Standard Deviation |
|---|---|---|---|
| GBRBM-AE | 5.53 | 6.35 | 0.12 |
| ANN | 7.53 | 7.8 | 0.37 |
| SVM | 10.37 | 13.48 | 0.63 |

Figures 9 and 10 depict the best results obtained from the SVM algorithm, and similarly, Figures 11 and 12 present the best results of the ANN method. Specifically, Figure 9 presents the value of the received signal obtained by the best SVM compared to the measurement the UAV took. The red dots represent the measurement values, while the blue line represents the SVM output. Although the SVM predicted value was close to the experimental value, it does not consider adequately near optimum values. Furthermore, Figure 10 depicts the histogram of the difference between the real and estimated values. Therefore, we may conclude that the SVM is well modeled but insufficient, as it has accentuated large errors. Table 4 indicates the best results obtained from GBRBM-AE, ANN, and SVM, and the indicators used are the mean absolute error (MAE), mean absolute percent error (MAPE), and the root mean squared error (RMSE).

**Table 4.** Comparative results of MAE, MAPE, and RMSE for different training algorithm methods.

| Algorithm | GBRBM-AE | ANN | SVM |
|---|---|---|---|
| MAE (dB) | 4.13 | 6.65 | 8.12 |
| MAPE (%) | 3.27 | 5.18 | 10.37 |
| RMSE (dB) | 7.17 | 8.32 | 12.63 |

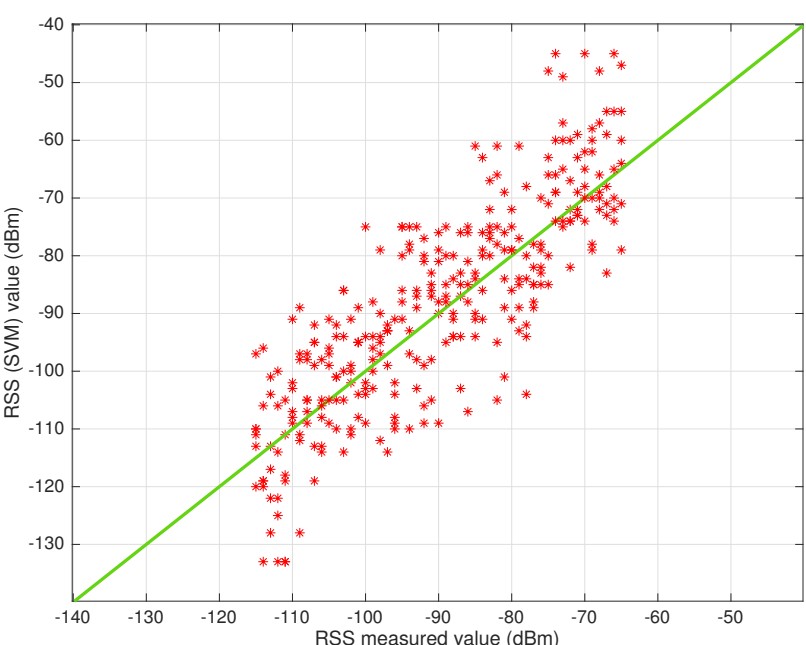

**Figure 9.** Difference in the RSS value of estimated (SVM) and real RSS values (dB).

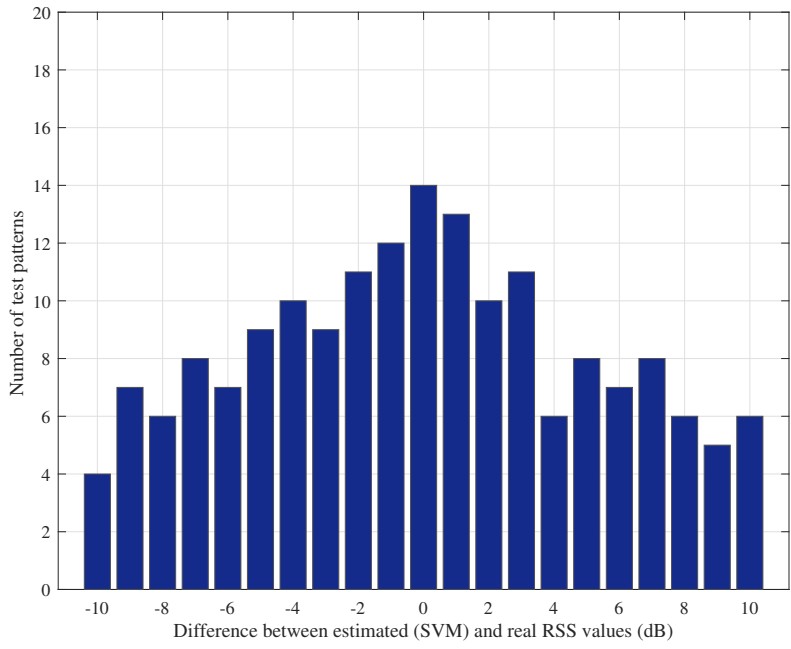

**Figure 10.** Statistical distribution of the difference in the RSS value of estimated (SVM) and real RSS values (dB).

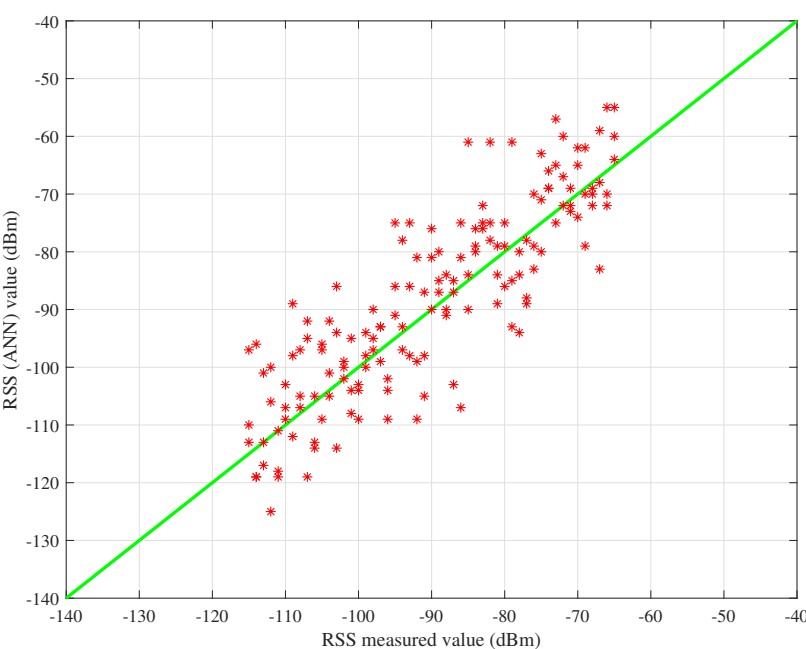

**Figure 11.** Difference in the RSS value of estimated (ANN) and real RSS values (dB).

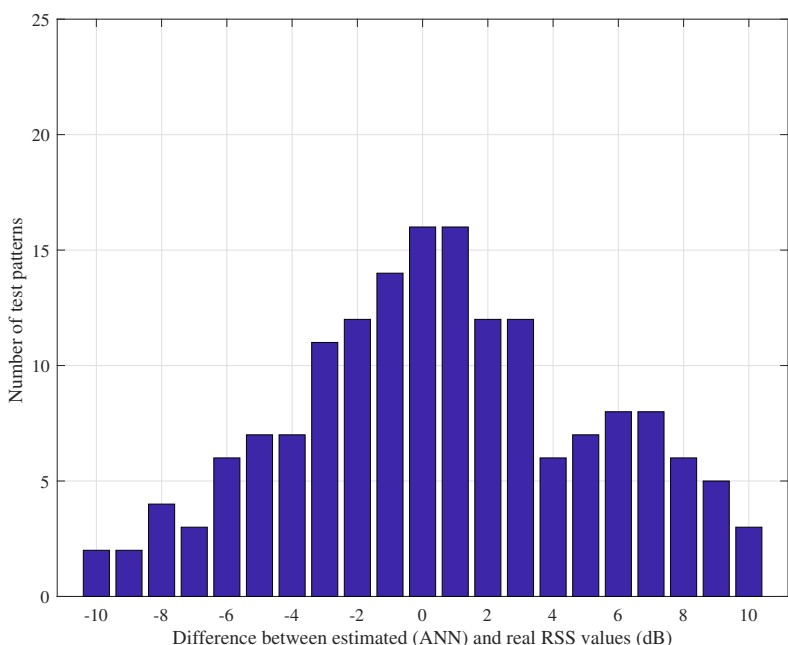

**Figure 12.** Statistical distribution of the difference in the RSS value of estimated (ANN) and real RSS values (dB).

In contrast, the results of ANN show a better performance, as can be seen in Figures 11 and 12. ANN performs better than SVM and adequately modeled the received signal power with better prediction accuracy. In addition, the best results were obtained with GBRBM-AE, as presented in Figures 13 and 14. Clearly, the values of the GBRBM-AE method are the best results compared to the previously considered algorithms. The green line represents the GBRBM-AE output, while the red dots depict real UAV measurements. In addition, we noticed that the predicted values are adequately close to the measurement values. The accuracy result of the proposed model is shown by the time-complexity analysis, which also shows how long it takes to train and validate the model in Table 5.

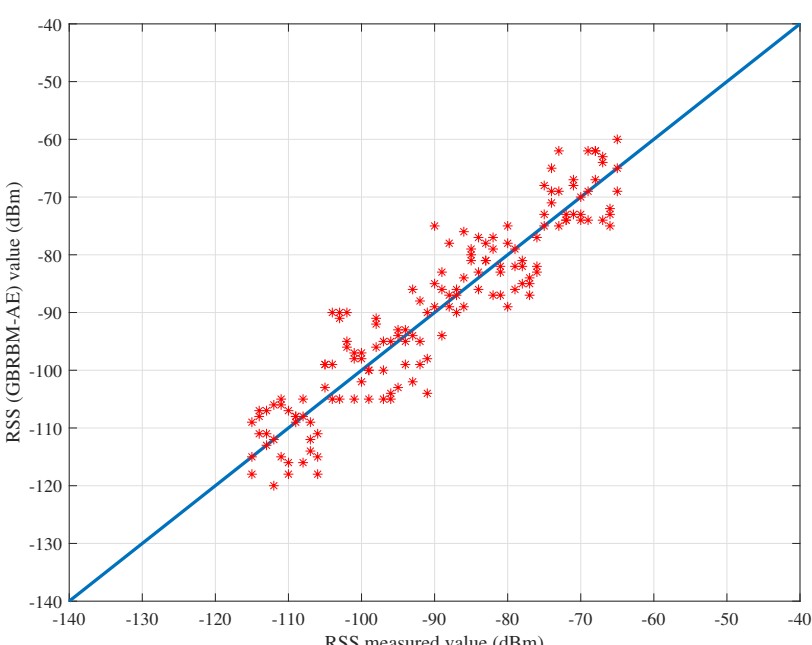

**Figure 13.** Difference in the RSS value of estimated (GBRBM-AE) and real RSS values (dB).

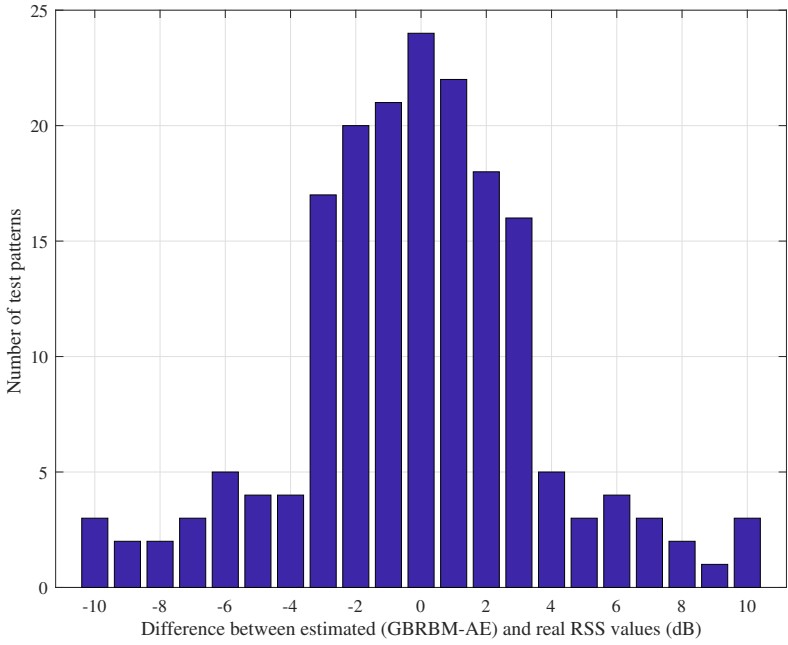

**Figure 14.** Statistical distribution of the difference in RSS value of estimated (GBRBM-AE) and real RSS values (dB).

**Table 5.** Time Cost.

| Algorithm | Time Cost (ms) |
| --- | --- |
| GBRBM-AE | 2.1 |
| ANN | 7 |
| SVM | 68 |

## 5. Conclusions

This paper proposed a framework based on GBRBMs integrated with an autoencoder-based DNN to estimate the received signal power at a UAV flying in a range of altitudes and connected to a cellular network. Real UAV signal power measurements are used for system training and validation. Although the proficiency of RBMs in exploring the latent features in an unsupervised manner, its training is challenging as the stochastic gradient tends to high variance and diverging behavior, and the learning rate must be manually set according to the RBMs trained structure. The problem of having meaningless hidden neurons during the training of RBMs is intense. To overcome these issues, a novel algorithm is proposed that uses an adaptive learning rate alongside with an enhanced gradient. The enhanced gradient is used to speed up the whole learning of hidden neurons, contrary to the traditional gradient decent. Furthermore, performance comparisons with SVM and ANN algorithms are provided to demonstrate the validity and gains of the proposed algorithm, where the obtained results have revealed that the GBRBM AE outperforms the other algorithms with a powerful optimization capability.

**Author Contributions:** Conceptualization, O.A.; Methodology, A.A.-G.; Formal analysis, A.Y.S. and H.A.-H. All authors have read and agreed to the published version of the manuscript.

**Funding:** This research received no external funding.

**Institutional Review Board Statement:** Not applicable.

**Informed Consent Statement:** Not applicable.

**Data Availability Statement:** Not applicable.

**Conflicts of Interest:** The authors declare no conflict of interest.

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
