# Peer review of "Channel Estimation for UAV Communication Systems Using Deep Neural Networks"

_drones, doi:10.3390/drones6110326_

Round 1

Reviewer 1 Report

The paper deals with the modelling of mobile channel for UAV based on measurements and ML. The paper is well written and organized.

This Reviewer has several concerns:

-       What are the main differences of this contribution with respect to [28] in terms of learning?. I mean, the application on [28] is different but the ML technique and learning process is similar, as far as I understood.

-       Introduction could be shortened. 4 pages for the introduction could be too much.

-       The description of GBRBM is not clear. For instance, encoder and decoder processes. Why are they needed? What are the goals? (Figure 4 need to be explained in more detail)

-       The measurement campaign is not clear and more details should be provided. For example, the authors specify the area of flying, the average height of drones but they do not say anything about the ground measurements points such as positions, heights, LoS or NLoS, equipment’s characteristics , etc. They do not say anything on the average number of measurements for each point, even the number of points, or the conditions of the ground area. I mean, is it a city? An open area, mountains?, forest? All these data should be specified.

-       The comparison with SVM and ANN should be better explained. For example, the authors should explain clearly if they have trained SVM and ANN with the same data set and variables. Also the number of iterations for the training in SVM, ANN and their proposal. Comparison against ray tracing would be appreciated.

-       Figures such as 8,10 and 12 are nice, but it would be better to evaluate the MSE.

-       Nothing is indicated on the complexity and real time. In fact, it seems to be offline. For a realistic application, these aspect should be analyzed.

I would suggest a different title because the title suggest a measurement campaign and based on that campaign, they have proposed a model by using DNN. But the main contribution of the paper is not the measurement campaign or data, or even the model, but a comparison of a SVM, ANN and their proposal for RSS estimation. So, a title similar to: Using Deep Neural Networks for estimating the RSS on UAV communications.

Figure 3 is confused. According to the figure, it seems that there are different blocks. However, from one block to the next one, the output layer of previous block is the input layer for next blocks, right? The first block and the second are coherent in colors (same color for output layer and next input layer), but then, colors are incoherent, so it is not clear if the output layer of previous block is the input layer for next block. Especially when they talk about blocks. In my opinion, they should talk about layers instead to clarify everything. Or I am missing something?

Minor:

Use the definition of acronym the first time it is used such as GBRBM or others. Do not repeat the definition several times as UAV in the paper.

Line 261: a reference to a figure is missing

Line 271, 276, 281. Is intended to have InputInput and OutputOutput?

I would suggest in figure 9, that y axes instead of being the number of paths, to be the percentage to figure out better the behavior. Same for figure 11

Reference 3 from the authors seems to be erroneous. In the paper the reference talks on UAV in cities and the reference is about RIS for smart cities.

Reference 14 from the authors is incomplete.

Author Response

We sincerely thank the Reviewer for the considerable time and effort putting into the technical review of this paper. We have carefully revised the paper according to your feedback and dealt with them one by one as follows.

Reviewer 2 Report

This paper presents air-to ground results from a UAV exploiting adaptive deep neural networks. The method and concept is of interest but the manuscript requires major revisions.

Firtstly, the title is misleading. The air-to-ground results are obtained from simulations and not from measurements. Therefore, the authors should make the corresponding corrections. The same stands for the header of Section 3.

Secondly, additional information about the simulation study must be provided. For example is there a digital map embedded in Wireless Insite? What kind of area is included (e.g., urban, suburban)? Were the yielded data for line-of-sight (LOS) or non-LOS environment? Finally, the most important is to refer what kind of data are resolved from the simulation procedure and how many samples. THis is related with the Deep Neura Network and its input and output features. Which parameters are used to train the network and what is the output (path loss or received power). How did the authors used the specific input features. Did they checked the feature importance?

Thirdly, the exact hyperparameters of the GBRRM as well as the SVM and ANN networks muste be provided in detail and summarized in a new Table.

Finally, the authors should introduce specific error metrics to examine the models' appropriateness. The figures alone are not adequate. For example mean absolute error, standard deviation, rms error, mean absolute percentage error, and correlation. A summarizing Table have to be added.

Author Response

(The authors gave the same response as above.)

Reviewer 3 Report

This paper proposes to use GBRBM to predict the channel from ground BS to UAV. The topic considered in this paper is interesting and timely. However, the proposed scheme is not well justified. Please find my detailed comments as below:

The authors clearly distinguish the two application scenarios of UAV, i.e., the cellular-connected UAV and UAV-assisted communication. In the former, the UAV is a user connected to the cellular network, while the UAV serves as the BS in the latter scenario. Given the different roles of the UAV, the way of channel estimation should be different.  

The classical path loss model has been shown to well-suit the LoS scenario. Provided the controllable trajectory of the UAV, we should be able to predict the channel gain of the UAV with classical models. The advantage of machine learning approach is not clear.

Most of the content of this paper is about the GBRBM models, which has been discussed in the existing works. The channel model of the BS to UAV link is not mentioned. For effective beamforming in mmWave communication, the channel impulse response is required, which is much more than the channel gain. It is not clear whether the proposed approach learn the complete channel information.

For the simulation, only the estimated RSS value is compared, which is not sufficient for mmWave beamforming. Furthermore, the comparison with the classical model-based channel prediction is missing.

The presentation of this paper needs to be improved, e.g., Line 276, “InputInput”; Line 156, “Significantly In natural disasters…”. Line 261, “Fig ??”.

Author Response

(The authors gave the same response as above.)

Round 2

Reviewer 1 Report

The authors have acomplished with all my reccomentdations. 

However, there are still some unclear statements when describing the scenarios and the measurements. It is not clear if the measurements campaign has been carried out and how it has been carried out. I mean, the authors given more details on the simulations, but still not details on the measurements. Durantion, number of measurements for each point, conditions, place....

I think the authors must clarify what have been done, what data come from simulations, what from measurements and what a re the results.

Author Response

We sincerely thank the Reviewer for the considerable time and efforts putting into the technical review of this paper.

Reviewer 2 Report

The authors addressed the majority of the raised issues, however, the last request is still not answered.

Therefore as a minor comment is that the authors introduce and calculate specific error metrics apart from Figures 9 , 11 and 13. For example the mean absolute error, the standard deviation, rms error, mean absolute percentage error and correlation between the estimated and real RSS values. Then a new Table should summarize the results.

Author Response

(The authors gave the same response as above.)

Reviewer 3 Report

I don't have any further comments. 

Author Response

(The authors gave the same response as above.)
